# LEARNING GRAPH NEURAL NETWORK TOPOLOGIES

## ABSTRACT

Graph convolutional networks (GCNs) enable end-to-end learning on graph structured data. However, many works begin by assuming a given graph structure. As the ideal graph structure is often unknown, this limits applicability. To address this, we present a novel end-to-end differentiable graph-generator which builds the graph topology on the fly. Our module can be readily integrated into existing pipelines involving graph convolution operations, replacing the predetermined or existing adjacency matrix with one that is learned, and optimised, as part of the general objective. As such it is applicable to any GCN. We show that integrating our module into both node classification and trajectory prediction pipelines improves accuracy across a range of datasets and backbones.

## 1 INTRODUCTION

The success of Graph Neural Networks (GNNs) (Duvenaud et al., 2015; Bronstein et al., 2017; Monti et al., 2017), has led to a surge in the use of graph-based representation learning. GNNs provide an efficient framework to learn from graph-structured data, making them widely applicable in any domain where data can be represented as a relation or interaction system. They have been successfully applied in a wide range of tasks including particle physics (Choma et al., 2018), protein science (Gainza et al., 2020) and many others (Monti et al., 2019), (Stokes et al., 2020).

In a GNN, each node iteratively updates its state by interacting with its neighbors, typically through message passing. However, a fundamental limitation of such architectures is the assumption that the underlying graph is provided. While node or edge features may be updated during message passing, the graph topology remains fixed, and its choice may be suboptimal for various reasons. For instance, when classifying nodes on a citation network, an edge connecting nodes of different classes can diminish classification accuracy. These edges can degrade performance by causing irrelevant information to be propagated across the graph. When no graph is explicitly provided, one common practice is to generate a $k$-nearest neighbor ($k$-NN) graph. In such cases, $k$ is a hyperparameter and tuned to find the model with the best performance. For many applications, fixing $k$ is overly restrictive as the optimal choice of $k$ may vary for each node in the graph. While there has been an emergence of approaches which learn the graph structure for use in downstream GNNs (Zheng et al., 2020; Kazi et al., 2020; Kipf et al., 2018), all of them treat the node degree $k$ as a fixed hyperparameter.

We propose a general differentiable graph-generator (DGG) module for learning graph topology with or without an initial edge structure. This module can be placed within any graph convolutional network, and jointly optimized with the rest of the network's parameters, learning topologies which favor the downstream task without hyperparameter selection or indeed any additional training signal. The primary contributions of this paper are as follows:

1. We propose a novel, differentiable graph-generator (DGG) module which jointly optimizes both the neighbourhood size, and the edges that should belong to each neighbourhood. Note that existing approaches (Zheng et al., 2020; Kipf et al., 2018; Kazi et al., 2020) do not allow for learnable neighbourhood sizes.

2. Our DGG module is directly integrable into any pipeline involving graph convolutions, where either the given adjacency matrix is noisy, or is not explicitly provided and must be determined heuristically. In both cases, our DGG generates the adjacency matrix as part of the GNN training and can be trained end-to-end to optimize performance on the downstream task. Should a good graph structure be known, the generated adjacency matrix can be learned to remain close to it while optimizing performance.

3. To demonstrate the power of the approach, we integrate our DGG within a range of SOTA pipelines — without modification — across different datasets in trajectory prediction and node classification and demonstrate improvements in model accuracy.

## 2 RELATED WORK

**Graph Representation Learning:** GNNs (Bronstein et al., 2017) provide a powerful class of neural architectures for modelling data which can be represented as a set of nodes and relations (edges). Most use message-passing to build node representations by aggregating neighborhood information. A common formulation is the Graph Convolution Network (GCNs) which generalizes the convolution operation to graphs (Kipf & Welling, 2017; Defferrard et al., 2016; Wu et al., 2018; Hamilton et al., 2017). More recently, the Graph Attention Network (GAT) (Veličković et al., 2018) utilizes a self-attention mechanism to aggregate neighborhood information. However, these works assumed that the underlying graph structure is predetermined, with the graph convolutions learning features that describe preexisting nodes and edges. In contrast, we simultaneously learn the graph structure while using our generated adjacency matrix in downstream graph convolutions. The generated graph topology of our module is jointly optimized alongside other network parameters with feedback signals from the downstream task.

**Graph Structure Learning:** In many applications, the optimal graph is unknown, and a graph is constructed before training a GNN. One question to ask is: "Why isn't a fully-connected graph suitable?" Constructing adjacency matrices weighted by distance or even an attention mechanism (Veličković et al., 2018) over a fully-connected graph incorporates many task-irrelevant edges, even if their weights are small. While an attention mechanism can zero these out — i.e., discover a subgraph within the complete graph — discovering this subgraph is challenging given the combinatorial complexity of graphs. A common remedy is to sparsify a complete graph by selecting the $k$-nearest neighbors ($k$-NN). Although this can prevent the propagation of irrelevant information between nodes, the topology of the constructed graph may have no relation to the downstream task. Not only can irrelevant edges still exist, but pairs of relevant nodes may remain unconnected and can lead GCNs to learn representations with poor generalization (Zheng et al., 2020).

This limitation has led to works which learn a graph's structure within a deep learning framework. Some methods (Shi et al., 2019; Liu et al., 2020) take a fixed adjacency matrix as input and then learn a residual mask over it. Since these methods directly optimize the residual adjacency by treating each element as a learnable parameter, the learned adjacency matrix is not linked to the representation space and only works in tasks where the training nodes are the same as that at test time. To overcome this, recent approaches (Zheng et al., 2020; Kipf et al., 2018; Luo et al., 2021; Kazi et al., 2020) generate a graph structure by sampling from discrete distributions. As discrete sampling is not directly optimizable using gradient descent, these methods use the Gumbel-Softmax reparameterization trick (Jang et al., 2016) to generate differentiable graph samples. The Gumbel-Softmax approximates an argmax over the edges for each node, and sampling in these approaches is typically performed $k$ times to obtain the top-$k$ edges. Here, $k$ is a specified hyperparameter that controls the node degree for the entire graph/dataset. Unlike these works, we generate edge samples by selecting the top-$k$ in a differentiable manner, where we learn a distribution over the edges *and* over the node degree $k$. This allows the neighborhood and its size to be individually selected for each node. Additionally, a known 'ideal' graph structure can be used as intermediate supervision to further constrain the latent space.

## 3 METHOD

In this section, we provide details of our differentiable graph generation (DGG) module. We begin with notation and the statistical learning framework guiding its design, before describing the module, and how it is combined with graph convolutional backbone architectures.

**Notation** We represent a graph of $N$ nodes as $G = (V, E)$: where $V$ is the set of nodes or vertices, and $E$ the edge set. A graph's structure can be described by its adjacency matrix $A$, with $a_{ij} = 1$ if an edge connects nodes $i$ and $j$ and $a_{ij} = 0$ otherwise. This binary adjacency matrix $A$ is directed, and potentially asymmetrical.

**Problem definition.** We reformulate the baseline prediction task based on a fixed graph with an adaptive variant where the graph is learned. Typically, such baseline tasks make learned predictions $Y$ given a set of input features $X$ and a graph structure $A$ of node degree $k$:

$$Y = Q_\phi(X, A), \tag{1}$$

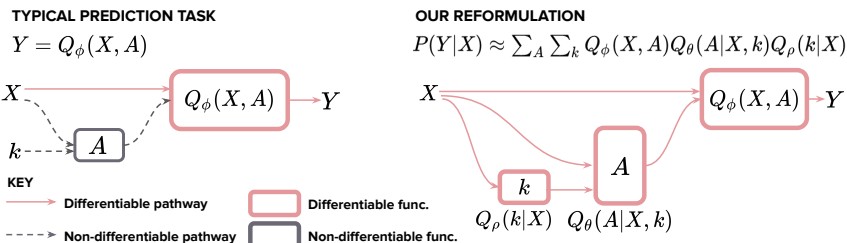

Figure 1: (Left) A typical prediction task using graphs $Y = Q_\phi(X, A)$ where $A$ and $k$ are predetermined. (Right) Our reformulation $P(Y|X) \approx \sum_A \sum_k Q_\phi(X, A)Q_\theta(A|X, k)Q_\rho(k|X)$ which learns a distribution over $A$ and $k$ alongside the downstream task.

where $Q_\phi$ is an end-to-end neural network parameterized by learnable weights $\phi$. These formulations require a predetermined graph-structure $A$, typically based on choice of node degree $k$, and take $A$ as additional input to the model. In contrast, we *learn* both $A$ and $k$ in an end-to-end manner, and use them to make predictions $Y$. As graphs are inherently binary, with edges either present or absent, they are not directly optimizable using gradient descent. Instead, we consider a distribution of graphs, $\mathcal{G}$, which then induce a distribution of labels, $\mathcal{Y}$, in the downstream task. This distribution takes the factorized form:

$$P(Y|X) = \sum_{A \in \mathcal{G}} \sum_{k \in \mathbb{N}^{|V|}} Q_\phi(X, A)P(A|X, k)P(k|X), \qquad (2)$$

where $P(k|X)$ is the distribution of node degree $k$ given $X$ (i.e., the choice of $k$ in $k-$NN), $P(A|X, k)$ the distribution of graph structures $A$ conditioned on the learned $k$ and input $X$, and $P(Y|X)$ is the downstream distribution of labels conditioned on data $X$. For clarity, the adjacency $A$ represents a subgraph of a complete graph over $X$, and $k$ is a multidimensional variable controlling the number of top-$k$ neighbors for each node individually. To avoid learning individual probabilities for each possible graph $A$ in an exponential state space, we further assume that $P(A|X, k)$ has a factorized distribution where each neighborhood is sampled independently, i.e. $P(A|X, k) = \prod_{i \in V} P(a_i|X, k)$.

We approximate the distributions over adjacencies $A$ and $k$ with tractable functions:

$$P(Y|X) \approx \sum_A \sum_k Q_\phi(X, A)Q_\theta(A|X, k)Q_\rho(k|X), \qquad (3)$$

where $Q_\theta$ and $Q_\rho$ are functions parameterized by $\theta$ and $\rho$ to approximate $P(A|X, k)$ and $P(k|X)$, respectively. In Fig. 1, we illustrate the functions of our method compared to the typical prediction task in Eq. 1.

Using this formulation, we train the entire system end-to-end to minimize the expected loss when sampling $Y$. This can be efficiently performed using stochastic gradient descent. In the forward pass, we first sample a subgraph/set of nodes $X$ from the space of datapoints, and conditioning on $X$ we sample $A$ and compute the associated label $Y$. When computing the gradient step, we update $Q_\phi(X, A)$ as normal and update the distributions using two standard reparametrization tricks: one for discrete variables (Jang et al., 2016) such that $Q_\theta(A|X, k)$ can generate differentiable graph samples $A'$, and another for continuous variables (Kingma & Welling, 2013) of $k'$ drawn from $Q_\rho(k|X)$:

$$P(Y|X) \approx \sum_{A'} \sum_{k'} Q_\phi(X, A'), \text{where } A' \sim Q_\theta(A|X, k') \text{ and } k' \sim Q_\rho(k|X). \qquad (4)$$

As both the graph structure $A'$ and variable $k'$ samplers are differentiable, our DGG module can be readily integrated into pipelines involving graph convolutions and jointly trained end-to-end.

### 3.1 Differentiable Graph Generation

Our differentiable graph-generator (DGG) takes a set of nodes $V = \{v_1, ..., v_N\}$ with $d$-dimensional features $\mathbf{X} \in \mathbb{R}^{N \times d}$ and generates an asymmetric adjacency matrix $\mathbf{A} \in \mathbb{R}^{N \times N}$. This adjacency matrix can be used directly in any downstream graph convolution operation (see Module Instatiation below). As illustrated by Fig. 2, the DGG module consists of four components:

1. **Encoder:** this component projects the input node features $\mathbf{X} \in \mathbb{R}^{N \times d}$ to a latent representation $\hat{\mathbf{X}} \in \mathbb{R}^{N \times d'}$, and forms the primary representation space for the model.

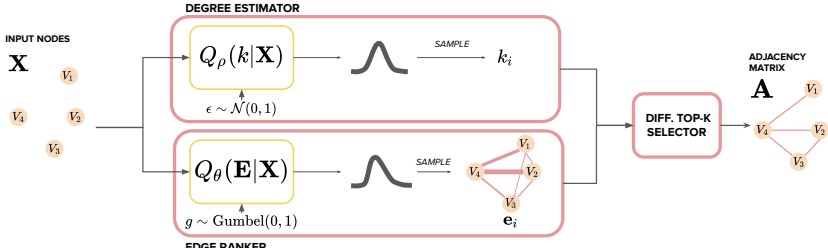

Figure 2: Our differentiable graph-generator (DGG) takes input nodes $\mathbf{X}$ and generates an adjacency matrix $\mathbf{A}$. It consists of: (1) **Degree-estimator**: generates samples of $k_i$ for each node, (2) **Edge-ranker**: generates edge samples $\mathbf{e}_i$ for each node and (3) **Top-k selector**: takes $k_i$ and edge samples $\mathbf{e}_i$ and selects top-k elements in a differentiable manner to output a final adjacency $\mathbf{A}$.

2. **Edge ranking**: this takes the latent node features $\hat{\mathbf{X}} \in \mathbb{R}^{N \times d'}$ and generates a matrix representing a stochastic ordering of edges $\mathbf{E} \in \mathbb{R}^{N \times N}$ drawn from a learned distribution over the edge-probabilities ($A' \sim Q_\theta(A|X, k')$ from Eq. 4).

3. **Degree estimation**: this component estimates the number of neighbors each individual node is connected to. It takes as input the latent node features $\hat{\mathbf{X}} \in \mathbb{R}^{N \times d'}$ and generates random samples $k \in \mathbb{R}^N$ drawn from a learned distribution over the node degree ($k' \sim Q_\rho(k|X)$ from Eq. 4).

4. **Differentiable top-$k$ edge selector**: takes $k$ and the edge-samples $e$ and performs a soft thresholding that probabilistically selects the most important elements, based on the output of the Edge-ranking to output an adjacency matrix $\mathbf{A} \in \mathbb{R}^{N \times N}$.

We describe each component below.

**Encoder.** We construct a single latent space from the input node features, and use it for edge ranking and degree estimation. We first map input node features $\mathbf{X} \in \mathbb{R}^{N \times d}$ into latent features $\hat{\mathbf{X}} \in \mathbb{R}^{N \times d'}$ using a multi-layer perceptron (MLP) $f_\phi$ with weights $\phi$: $\hat{\mathbf{X}} = f_\phi(\mathbf{X})$. These latent features form the input for the rest of the module. Furthermore, they are output by the DGG and passed to the GCN downstream to prevent vanishing gradients.

**Edge ranking.** The edge ranking returns an implicit distribution of edge orderings, from which we sample the neighborhood for each node. For each node $v_i$, it draws a set of scores $\mathbf{e}_i = \{e_{ij}\}_j^N$ quantifying its relevance to all nodes $v_j \in V$, including itself. To generate differentiable edge samples $\mathbf{e}_i$, we use the Gumbel-Softmax (Jang et al., 2016).

Without loss of generality, we focus on a single node $v_i \in V$, with latent features $\hat{\mathbf{x}}_i \in \mathbb{R}^d$. We implement the approximation function $Q_\theta(A|X, k)$ of the Edge-ranker as follows:

1. Using latent node features $\hat{\mathbf{x}}_i \in \hat{\mathbf{X}}$, calculate pairwise edge probabilities $\mathbf{p}_i \in \mathbb{R}^N$ between pairs of auxiliary node features $(\hat{\mathbf{x}}_i, \hat{\mathbf{x}}_j)$:

$$\mathbf{p}_i = \{\exp(-||\hat{\mathbf{c}}_{ij}||_1)|\forall j \in N\}, \tag{5}$$

where $\hat{\mathbf{c}}_{ij} = \hat{\mathbf{x}}_i - \hat{\mathbf{x}}_j$ is the difference between node features. Each element $p_{ij} \in \mathbf{p}_i$ represents a similarity measure between the latent features of node $v_i$ and $v_j$. In practice, any distance measure can be used here, including learnable approaches.

2. Using Gumbel-Softmax over the edge probabilities $\mathbf{p}_i \in \mathbb{R}^N$, we generate differentiable samples $\mathbf{e}_i \in \mathbb{R}^N$ with Gumbel noise $g$:

$$\mathbf{e}_i = \left\{ \frac{\exp((\log(p_{ij}) + g_i) + \tau)}{\sum_j \exp((\log(p_{ij}) + g_i) + \tau)} \middle| \forall j \in N \right\}, g \sim \text{Gumbel}(0, 1) \tag{6}$$

where $\tau$ is a temperature hyperparameter controlling the interpolation between a discrete one-hot categorical distribution and a continuous categorical density. When $\tau \to 0$, the edge energies $e_{ij} \in \mathbf{e}_i$ approach a degenerate distribution. The temperature $\tau$ is important for inducing sparsity, but given the exponential function, this results in a single element in $\mathbf{e}_i$ given much more weighting than the rest, i.e., it approaches a one-hot argmax over $\mathbf{e}_i$. Instead, we want a variable number of edges to be given higher importance, and others to be close to zero. Hence, we select a higher temperature and use the top-$k$ selection procedure (detailed below) to induce sparsity. This has the added benefit of avoiding the high-variance gradients induced by lower temperatures.

**Degree estimation.** A key limitation of existing graph generation methods (Kazi et al., 2020; Kipf et al., 2018; Zheng et al., 2020) is their use of a fixed node degree $k$ across the entire graph. This can be suboptimal as mentioned previously. In our approach, rather than fixing $k$ for the entire graph, we sample it per node from a learned distribution. Focusing on a single node as before, the approximation function $Q_\rho(k|X)$ of the Degree-estimator works as follows:

1. We approximate the distribution of latent node features $\hat{x}_i \in \mathbb{R}^d$ following the a VAE-like formulation (Kingma & Welling, 2013). We encode its mean $\boldsymbol{\mu}_i \in \mathbb{R}^d$ and variance $\boldsymbol{\sigma}_i \in \mathbb{R}^d$ using two MLPs $M_\rho$ and $S_\rho$, and then reparametrize with noise $\epsilon$ to obtain latent variable $\mathbf{z}_i \in \mathbb{R}^d$:

$$\boldsymbol{\mu}_i, \boldsymbol{\sigma}_i = M_\rho(\hat{\mathbf{x}}_i), S_\rho(\hat{\mathbf{x}}_i),$$
$$\mathbf{z}_i = \boldsymbol{\mu}_i + \boldsymbol{\epsilon}_i \boldsymbol{\sigma}_i, \epsilon_i \sim \mathcal{N}(0, 1). \tag{7}$$

2. Finally, we concatenate each latent variable $\mathbf{z}_i \in \mathbb{R}^d$ with the L1-norm of the edge samples $\mathbf{h}_i = ||\mathbf{e}_{ij}||_1$ and decode it into a scalar $k_i \in \mathbb{R}$ with another MLP $D_\rho$, representing a continuous relaxation of the neighborhood size for node $v_i$:

$$k_i = D_\rho(\mathbf{z}_i + \mathbf{h}_i). \tag{8}$$

Since $\mathbf{h}_i$ is a summation of a node's edge probabilities, it can be understood as representing an initial estimate of the node degree which is then improved by combining with a second node representation $\mathbf{z}_i$ based entirely on the node's features. Using the edge samples to estimate the node degree links these representation spaces back to the primary latent space of node features $\hat{\mathbf{X}}$.

**Top-$k$ Edge-Selector.** Having sampled edge weights, and node degrees $k$, this function selects the top-$k$ edges for each node. The top-$k$ operation, i.e. finding the indices corresponding to the $k$ largest elements in a set of values, is a piecewise constant function and cannot be directly used in gradient-based optimization. Previous work (Xie et al., 2020) frames the top-$k$ operation as an optimal transport problem, providing a smoothed top-$k$ approximator. However, as their function is only defined for discrete values of $k$ it cannot be optimized with gradient descent. As an alternative that is differentiable with respect to $k$, we relax the discrete constraint on $k$, and instead use it to control the $x$-axis value of the inflection point on a smoothed-Heaviside function (Fig. 3). For a node $v_i \in V$, of smoothed degree $k_i \in \mathbb{R}$ and edges $\mathbf{e}_i \in \mathbb{R}^N$, our Top-$k$ Edge Selector outputs an adjacency vector $\mathbf{a}_i \in \mathbb{R}^N$ where the $k$ largest elements from $\mathbf{e}_i$ are close to 1, and the rest close to 0. Focusing on a single node $v_i$ as before, the implementation is as follows:

1. Draw 1D input points $\mathbf{d}_i = \{1, ..., N\}$ where $N$ is the number of nodes in $V$.

2. Pass $\mathbf{d}_i$ through a hyperbolic tangent (tanh) which serves as a smooth approximation of the Heaviside function:

$$\mathbf{h}_i = 1 - 0.5 * \left\{ 1 + \tanh(\lambda^{-1} d_i - \lambda^{-1} k_i) \right\}, \tag{9}$$

here $\lambda > 0$ is a temperature parameter controlling the gradient of the function's inflection point. As $\lambda \to 0$, the smooth function approaches the Heaviside step function. The first-$k$ values in $\mathbf{h}_i = \{h_{ij}\}_j^N$ will now be closer to 1, while the rest closer to 0.

3. Finally, for each node $i$ we sort its edge-energies $\mathbf{e}_i = \{e_{ij}\}_j^N$ in descending order, multiply by $\mathbf{h}_i = \{h_{ij}\}_j^N$ and then restore the original order to obtain the final adjacency vector $\mathbf{a}_i = \{a_{ij}\}_j^N$. Stacking $\mathbf{a}_i$ over all nodes $v_i \in V$ then creates the final adjacency matrix $\mathbf{A} \in \mathbb{R}^{N \times N}$.

**Straight through Top-$k$ Edge Selector.** To make our final adjacency matrix $\mathbf{A} \in \mathbb{R}^{N \times N}$ discrete, we follow the trick used in the Straight-Through Gumbel Softmax (Jang et al., 2016): we output the discretized version of $\mathbf{A}$ in the forward pass and the continuous version in the backwards pass. For the discretized version in the forward pass, we simply replace the smooth-Heaviside function in Eq. 9 with a step function.

**Module Instantiation:** The DGG module can be easily combined with any graph convolution operation. A typical graph convolution (Kipf & Welling, 2017) is defined as follows: $\mathbf{X}' = \hat{\mathbf{D}}^{-1/2} \hat{\mathbf{A}} \hat{\mathbf{D}}^{-1/2} \mathbf{X} \boldsymbol{\Theta}$. Here, $\hat{\mathbf{A}} = \mathbf{A} + \mathbf{I}$ denotes the adjacency matrix with inserted self-loops, $\hat{\mathbf{D}}$ its diagonal degree matrix and $\boldsymbol{\Theta}$ its weights. To use this graph convolution with the DGG, we simply use our module to generate the adjacency matrix $\hat{\mathbf{A}}$.

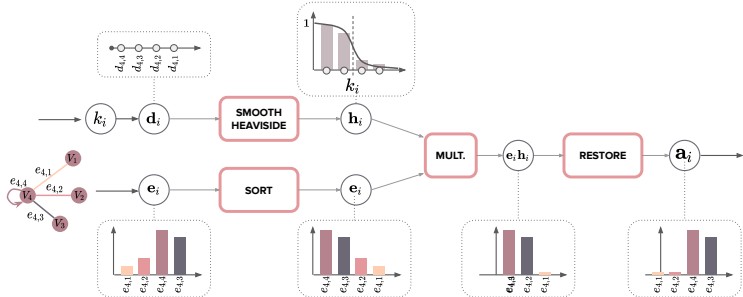

Figure 3: The differentiable Top-$k$ Edge Selector. This component uses the node degree $k_i$ output by the Degree Estimator to control the inflection point on a smooth-Heaviside function and uses it to select the top edges from $\mathbf{e}_i$ output by the Edge Ranker. This produces an adjacency vector $\mathbf{a}_i$ for each node, and stacking $\mathbf{a}_i$ across all nodes produces the final adjacency matrix $\mathbf{A}$.

## 4 EXPERIMENTS

We evaluate our DGG on node classification and trajectory prediction tasks. We chose these tasks as the former has an input graph structure, while the latter does not, demonstrating the flexibility of our module. Furthermore, we integrate our module into a variety of GCN baselines, not only for their state-of-the-art (SOTA) performance, but as they all use different graph convolution operations.

### 4.1 NODE CLASSIFICATION

Beginning with node classification, we conduct ablations examining the behaviour of different parts of the DGG, followed by comparisons to other state-of-the-art graph-topology learning approaches. In the appendix we include experiments investigating the effect of the DGG on downstream models under the addition of noisy edges to input graphs. We perform these experiments under both transductive and inductive scenarios, as well as semi-supervised and fully-supervised settings.

**Transductive datasets.** We evaluate on the three citation benchmark datasets Cora, Citeseer and Pubmed (NLM 2022) introduced by Yang et al. (2016). Each citation graph contains nodes corresponding to documents and edges indicating citations. Node features are bag-of-words features and node labels are categorized by topic. We follow the semi-supervised setting in Yang et al. (2016) and Kipf & Welling (2017) along with their train/test splits. **Inductive datasets.** In an inductive setting, we evaluate our approach on three datasets: 1. Flickr (Zeng et al., 2020) — categorizing images based on their descriptions; 2. Reddit (Zeng et al., 2020) — predicting the communities of online posts from user comments. 3. PPI (Hamilton et al., 2017) — classifying protein-protein interactions. Further dataset details can be found in the appendix.

**Baselines.** We integrate our DGG into four representative GNN backbones: GCN (Kipf & Welling, 2017), GraphSage (Hamilton et al., 2017), GAT (Veličković et al., 2018) and GCNII (Chen et al., 2020). On these backbones, we compare the effectiveness of DGG against other state-of-the-art graph sampling and sparsification methods: DropEdge (Rong et al., 2020), NeuralSparse (Zheng et al., 2020) and PTDNet (Luo et al., 2021).

**Implementation details.** We integrate our DGG into the official publicly available code of all baselines, without architectural modification. Models are trained and evaluated as their original implementation. See appendix for further details.

**Training details.** A node classification model partitions the latent space of node embeddings into separate classes. However, when message-passing, there is one phenomenon of the input graph that can limit classification accuracy: two nodes with different classes but similar features and an edge connecting them. Classifying these nodes is challenging as their feature similarity can be compounded by passing messages between them. The goal of the DGG is to move such nodes apart in the latent space such that there is no edge and communication between them. However, traversing the loss landscape from the initial random initialization of the network to one where the model is able to discriminate between these nodes can take several iterations using only the downstream classification loss. To speed up training, we add an intermediate loss to further partition the latent space. We do this by supervising the adjacency matrix generated by the DGG to remove all edges between classes and only maintain those within a class. We then anneal this loss over the training cycle, eventually leaving only the downstream classification loss. We provide more details in the appendix.

### 4.1.1 ABLATIONS

In Table 1, we explore the effect of disabling different components of our DGG module when integrated into a GCN (Kipf & Welling, 2017) for node classification: 1. *DGG without Degree Estimation and Differentiable Top-$k$ Edge Selection* — we remove the Degree Estimator and instead fixe $k$ to select the top-$k$ stochastically ordered edges. 2. *DGG with deterministic Edge Ranking* — we remove the noise in Eq. 6 of the Edge Ranker. 3. *DGG with deterministic Degree Estimation* — we remove the noise in Eq. 7 of the Degree Estimator. We perform this under the transductive, semi-supervised setting on Cora and omit the annealed intermediate loss during training.

Table 1: Ablation study. DGG integrated into a GCN for node classification on Cora.

| Model | Accuracy |
|---|---|
| Fixed node degree, k = {1, 5, 10, 100} | {49.7, 78.9, 55.0, 37.0} |
| With deterministic Edge Ranking and Degree Estimation | 82.4 |
| With deterministic Edge Ranking | 82.7 |
| With deterministic Degree Estimation | 82.8 |
| DGG | **83.2** |

Table 1 shows the benefit of learning a distribution over the node degree. When learning it deterministically, the accuracy decreases by 0.5%. This becomes significantly worse when the node degree is fixed for the entire graph rather than learned per node. Note also, the sensitivity with respect to choice of $k$. A fixed node degree of $k = 10$ or $k = 1$ reduces accuracy by almost 30% vs a graph of 5. This is due to the graph convolution operation: as it has no adaptive weighting mechanism for a node's neighborhood, each of the neighbors is given the same weight. Naturally, this leads to information sharing between unrelated nodes, reducing the quality of node representation after message-passing. In contrast, by learning a distribution over the node degree we are able to select only the most relevant neighbors, even though these are then weighted equally in the graph convolution. Finally, the inclusion of noise in any of the DGG components does increase accuracy, but only by about 0.5% — demonstrating both its benefit and the robustness of the DGG without it.

### 4.1.2 COMPARISON TO STATE-OF-THE-ART

In Table 2 we compare our method to DropEdge (Rong et al., 2020)), which randomly sparsifies the input graph, and those which learn better graph structures (NeuralSparse (Zheng et al., 2020) and PTDNet Luo et al. (2021)). For fair comparison with the literature, we present two versions of our method: DGG-wl trained with the downstream loss only and DGG* trained with both the downstream and intermediate loss.

DGG improves performance over the original model across all baselines and datasets. Against other approaches, DGG-wl generally outperforms the state-of-the-art NeuralSparse and PTDNet-wl (both trained with only the downstream loss). The accuracy difference can be attributed to our method for modelling sparsity, which explicitly lets each node to select the size of its neighborhood based on the downstream training signal. This training signal helps partition the node representation space, while the estimated node-degree additionally prevents communication between distant nodes. Although PTDNet-wl does this implicitly through its attention mechanism, discovering this sparse subgraph of the input graph is challenging given its complexity. NeuralSparse on the other hand selects $k$ for its entire generated subgraph, which is both suboptimal and requires additional hyperparameter tuning.

Comparing methods which enforce additional constraints on the adjacency matrix, DGG* demonstrates larger accuracy gains than PTDNet*. PTDNet* regularizes its adjacency matrix to be of low-rank, as previous work (Savas & Dhillon, 2011) has shown that the rank of an adjacency matrix can reflect the number of clusters. This regularizer reasons about the graph's topology globally. While this may aid generalization, the accuracy difference may then be attributed to our intermediate loss providing stronger signals to discriminate between nodes with similar features but different classes (and therefore remove the edges between them). Furthermore, their regularizer uses the sum of the top-$k$ singular values during training, where $k$ again is a hyperparameter tuned to each dataset individually. Our method requires no additional parameters to be chosen.

We observe that there may be cases where tuning the node degree $k$ can assist accuracy, as seen by NeuralSparse's performance on Reddit. However this requires a hyperparameter search over Reddit's large graph, and is ultimately outperformed when intermediate supervision is applied to the DGG's adjacency matrix (shown by the DGG* method).

Table 2: Semi-supervised node classification compared to other SOTA graph-topology learning methods. We compare against prior methods reported in (Luo et al., 2021; Zheng et al., 2020; Chen et al., 2020), using the official results where available.

| Backbone | Method | Cora | Citeseer | Pubmed | Reddit | PPI |
|----------|--------|------|----------|--------|--------|-----|
| GCN | Original | 81.1 | 70.3 | 79.0 | 92.2 | 53.2 |
| | DropEdge | 80.9 | 72.2 | 78.5 | 96.1 | 54.8 |
| | NeuralSparse | 82.1 | 71.5 | 78.8 | **96.6** | 65.1 |
| | PTDNet-wl | 82.4 | 71.7 | 79.1 | - | 75.2 |
| | DGG-wl | **82.7** | **72.4** | **80.1** | 96.5 | **76.7** |
| | PTDNet-wl + low rank | 82.8 | 72.7 | 79.8 | - | 80.3 |
| | DGG* | 83.9 | 74.9 | 83.9 | 97.2 | 81.5 |
| GraphSage | Original | 79.2 | 67.6 | 76.7 | 93.8 | 61.8 |
| | DropEdge | 78.7 | 67.0 | 74.8 | 96.3 | 61.0 |
| | NeuralSparse | 79.3 | 67.4 | 75.1 | **96.7** | 62.6 |
| | PTDNet-wl | **79.4** | 67.8 | 77.0 | - | 64.5 |
| | DGG-wl | **79.4** | **68.2** | **77.6** | **96.7** | **65.1** |
| | PTDNet-wl + low rank | 80.3 | 67.9 | 77.1 | - | 64.8 |
| | DGG* | 80.4 | 70.6 | 80.1 | 96.9 | 67.3 |
| GAT | Original | 83.0 | 72.1 | 79.0 | - | 97.3 |
| | DropEdge | 83.2 | 70.9 | 77.9 | - | 85.1 |
| | NeuralSparse | 83.4 | 72.4 | 78.0 | - | 92.1 |
| | PTDNet-wl | 83.7 | 72.3 | 79.2 | - | **97.8** |
| | DGG-wl | **84.2** | **73.0** | **79.5** | - | 97.3 |
| | PTDNet-wl + low rank | 84.4 | 73.7 | 79.3 | - | **98.0** |
| | DGG* | 85.1 | 76.3 | 81.8 | - | 97.5 |
| GCNII | Original | 85.3 | 73.2 | 80.2 | - | 99.5 |
| | DropEdge | 84.9 | 73.4 | 79.4 | - | 99.0 |
| | DGG-wl | 86.8 | 74.4 | 81.2 | - | 99.5 |
| | DGG* | **87.7** | **75.8** | **81.9** | - | **99.7** |

Table 3: Adjacency matrix constraints: our intermediate annealed loss vs. PTDNet's low rank regularizer Luo et al. (2021) for semi-supervised node classification with a GCN backbone.

| Method | Cora | Citeseer | Pubmed | Reddit | PPI |
|--------|------|----------|--------|--------|-----|
| Ours-wl | 82.7 | 72.4 | 80.1 | 96.1 | 76.7 |
| Ours-wl + low rank | 83.3 | 73.1 | 81.0 | 96.1 | 80.6 |
| Ours-wl + int. loss | **83.9** | **74.9** | **83.9** | 97.2 | 81.5 |
| Ours-wl + int. loss + low rank | 84.0 | 75.1 | 84.2 | 97.2 | 81.8 |

Finally in Table 3 we compare the low-rank constraint of PTDNet with our intermediate annealed loss. Our intermediate loss ('Ours-wl + int. loss') outperforms the low-rank constraint ('Ours-wl + low rank'). However, using both constraints ('Ours-wl + int. loss + low rank') increases classification accuracy further, suggesting the edges removed by both methods are complementary.

### 4.2 TRAJECTORY PREDICTION

We consider four datasets covering a range of scenarios from baseketball to crowded urban environments. On each, we integrate our DGG into a state-of-the-art trajectory prediction method and compare results to another state-of-the-art graph-topology learning approach DGM (Kazi et al., 2020).

**Datasets.** We evaluate on four trajectory prediction benchmarks. 1. **ETH** (Pellegrini et al., 2009) **and UCY** (Lerner et al., 2007) — 5 subsets of widely used real-world pedestrian trajectories . 2. **STATS SportVU** (SportVU) — multiple NBA seasons tracking trajectories of basketball players over a game. **Stanford Drone Dataset (SDD)** (Robicquet et al., 2016) — top-down scenes across multiple areas at Stanford University, consisting of different agents from pedestrians to cars. Further details on these datasets can be found in the appendix.

**Baselines.** We integrate our DGG module into two state-of-the-art trajectory prediction pipelines: **Social-STGCNN** (Mohamed et al., 2020) built upon a spatio-temporal convolutional network using graphs to represent pedestrian trajectories and **DAGNet** (Monti et al., 2021) built upon a VAE

Table 4: ADE/FDE metrics on the ETH & UCY datasets using Social-STGCNN. For DGM, $k = 2$.

| Dataset | Original | | DGM | | DGM Gain | | Ours | | Our Gain | |
|---|---|---|---|---|---|---|---|---|---|---|
| | ADE↓ | FDE↓ | ADE↓ | FDE↓ | ADE↑ | FDE↑ | ADE↓ | FDE↓ | ADE↑ | FDE↑ |
| ETH | 0.64 | 1.11 | 0.62 | 1.04 | 2.4% | 6.4% | 0.60 | 0.89 | **6.3%** | **20%** |
| Hotel | 0.49 | 0.85 | 0.42 | 0.69 | 14.2% | 18.9% | 0.38 | 0.54 | **22.4%** | **36.5%** |
| Univ | 0.44 | 0.79 | 0.41 | 0.76 | 6.2% | 3.5% | 0.40 | 0.70 | **9.1%** | **11.3%** |
| Zara1 | 0.34 | 0.53 | 0.33 | 0.46 | 3.8% | 13.7% | 0.32 | 0.42 | **5.9%** | **20.8%** |
| Zara2 | 0.30 | 0.48 | 0.29 | 0.43 | 5.0% | 5.0% | 0.27 | 0.40 | **10.0%** | **16.7%** |
| Mean | 0.44 | 0.75 | 0.41 | 0.68 | 6.3% | 10.6% | 0.39 | 0.59 | **11.4%** | **21.3%** |

Table 5: ADE/FDE metrics on the SportVU Basketball dataset using DAGNet. For DGM, $k = 3$.

| Split | Team | Original | | DGM | | DGM Gain | | Ours | | Our Gain | |
|---|---|---|---|---|---|---|---|---|---|---|---|
| | | ADE | FDE | ADE | FDE | ADE | FDE | ADE | FDE | ADE | FDE |
| 10-40 | ATK | 2.74 | 4.29 | 2.75 | 4.30 | -0.4% | -0.2% | 2.62 | 4.11 | **4.4%** | **4.2%** |
| | DEF | 2.09 | 2.97 | 2.10 | 2.97 | -0.5% | -0.1% | 1.95 | 2.87 | **6.8%** | **3.3%** |
| 20-30 | ATK | 2.03 | 3.98 | 2.03 | 3.98 | 0.1% | 0.1% | 1.92 | 3.75 | **5.6%** | **5.8%** |
| | DEF | 1.53 | 3.07 | 1.53 | 3.06 | 0.2% | 0.3% | 1.24 | 2.56 | **18.7%** | **16.6%** |
| 40-10 | ATK | 0.81 | 1.71 | 0.80 | 1.69 | 1.3% | 0.9% | 0.70 | 1.48 | **13.6%** | **13.5%** |
| | DEF | 0.72 | 1.49 | 0.71 | 1.48 | 0.8% | 0.8% | 0.66 | 1.29 | **8.3%** | **13.4%** |
| Mean | | 1.65 | 2.92 | 1.7 | 2.9 | 0.3% | 0.3% | 1.51 | 2.68 | **9.6%** | **9.5%** |

backbone (Kingma & Welling, 2013) with a graph attention network modelling agent interactions across a fully-connected graph. Our DGG is placed within both networks to generate the adjacency matrix on the fly and forms part of its forward and backward pass. To integrate the DGG with DAGNet's attention mechanism, the adjacency generated is multiplied by the attention weights.

**Implementation details.** We integrate DGG into the publicly available code of each method, without any architectural modification. We use the same DGG hyperparameters as for node classification except the intermediate loss is disabled and the training signal is entirely from the downstream task. **Evaluation metrics.** Model performance is measured with Average Displacement Error (ADE) and Final Displacement Error (FDE). ADE measures the average Euclidean distance along the entire predicted trajectory, while the FDE is that of the last timestep only.

### 4.2.1 RESULTS

In Table 4, the integration of our DGG into Social-STGCNN reduces ADE/FDE compared to both the baseline and the integration of DGM. In Table 5 and 6 our DGG displays similar gains over DGM when integrated into DAGNet. First, this shows the benefit of inducing sparsity when message-passing over a distance weighted adjacency matrix like Social-STGCNN or even an attention-mechanism like DAGNet. The larger error reduction of our DGG compared to DGM may be attributed to DGM's use of a fixed node-degree $k$ across its learned graph. While this can prevent the propagation of irrelevant information across the graph in some cases, in others it might limit the context available to certain nodes. On the other hand, trying to discover the subgraph entirely through attention makes optimization a challenge. Instead, constraining the model by allowing each node to select its neighborhood and size eases optimization and can still be done entirely from the downstream training signal. We provide further qualitative analysis for these results in the appendix.

## 5 CONCLUSION

We have presented a novel approach for learning graph topologies, and shown how it obtains state-of-the-art performance across multiple baselines and datasets for node classification and trajectory prediction. The principal advantage of our approach is that it can be combined with any existing approach using graph convolutions on an automatically generated graph, such as a $k-$nearest neighbor graph.

Table 6: ADE/FDE metrics on the Stanford Drone Dataset using DAGNet. For DGM, $k = 2$.

| Split | Original | | DGM | | DGM Gain | | Ours | | Our Gain | |
|---|---|---|---|---|---|---|---|---|---|---|
| | ADE | FDE | ADE | FDE | ADE | FDE | ADE | FDE | ADE | FDE |
| 8-12 | 0.53 | 1.04 | 0.52 | 1.01 | 1.9% | 3.0% | 0.48 | 0.96 | **10.4%** | **8.3%** |

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

# A  NODE CLASSIFICATION EXPERIMENTS

## A.1  DATASET DETAILS

In Table 7 we provide dataset statistics on the node classification datasets used in this paper.

Table 7: Dataset statistics. 's' stands for single class classification and 'm' for multi-class.

| Dataset | Nodes | Edges | Features | Classes | Train / Val / Test |
|---------|-------|-------|----------|---------|--------------------|
| Cora | 2,708 | 5,429 | 1,433 | 7 (s) | 140 / 500 / 1,000 |
| Citeseer | 3,327 | 4,732 | 3,703 | 6 (s) | 120 / 500 / 1,000 |
| PubMed | 19,717 | 44,338 | 500 | 3 (s) | 100 / 500 / 1,000 |
| PPI | 56,944 | 81,8716 | 50 | 121 (m) | 44,906 / 6,514 / 5,524 |
| Flickr | 89,250 | 899,756 | 500 | 7 (s) | 44,760 / 22,312 / 22,312 |
| Reddit | 232,965 | 11,606,919 | 602 | 41 (s) | 153,756 / 23,295 / 55,911 |

## A.2  IMPLEMENTATION DETAILS

In our DGG, all MLPs use a signle fully-connected layer of dimension 64. In our DGG, all MLPs use a single fully-connected layer of dimension 64. We use a Gumbel-Softmax temperature of 2.5. When adding Gumbel noise to the edge log-probabilities, we do not add any to the self-loops (i.e. the diagonal of the edge probability matrix). During training, we keep the Gaussian and Gumbel noise on, but turn it off during inference. While in practice it can be left on, we found it does not significantly impact the results.

## A.3  TRAINING DETAILS

We train the entire network end-to-end using the classification loss from the downstream model and an annealed MSE loss on the adjacency matrix generated by the DGG:

$$\mathcal{L}_{\text{total}} = \mathcal{L}_{\text{class}} + \frac{\alpha}{M} \sum_{i=1}^{M} (y_i - \hat{y_i})^2, \tag{10}$$

where the first term is the classification loss from the downstream GCN, the second term is the MSE loss applied to every element $\hat{y_i}$ of the adjacency matrix $\mathbf{A} \in \hat{\mathbb{R}}^{N \times N}$ for which we have a node label, and $\alpha$ is loss weight. The model can be trained by annealing $\alpha$ smoothly or in a step-wise manner. In practice we keep $\alpha$ constant for the first 100 epochs and then set it to zero for the rest of the training schedule (where the total number of epochs is determined by schedule of the downstream GCN).

### A.4 ROBUSTNESS TO NOISE

We test the effect of the DGG when the input graph has random edges added across it. We do this by adding edges between previously unconnected nodes. Broadly, the results in Fig. 4 demonstrate the detrimental effects of noisy edges on classification accuracy, but the inclusion of the DGG can mitigate this. Interestingly, the state-of-the-art GCNII Chen et al. (2020) demonstrates the largest drops in accuracy as the noise increases, which may be attributed to the depth of their graph convolution layers. In such deeper message-passing models, the edge structure is even more significant, highlighting the importance of learning a structure that prevents the propagation of irrelevant information.

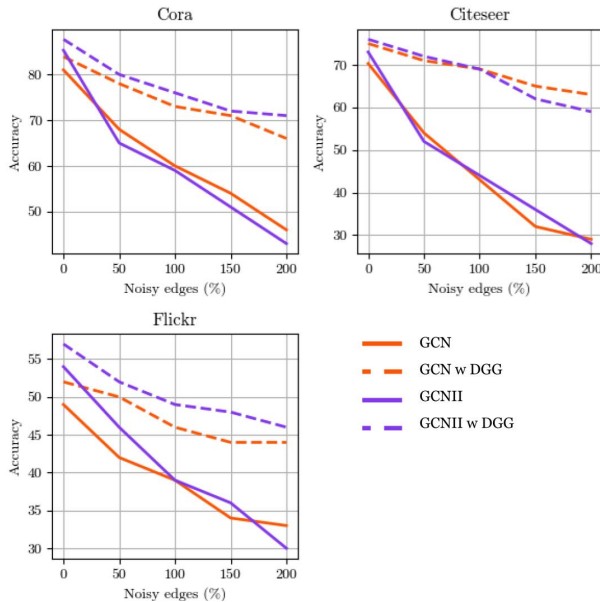

Figure 4: Node classification accuracy with noisy edges added to the input graph of different datasets.

## B TRAJECTORY PREDICTION EXPERIMENTS

### B.1 DATASET DETAILS

**ETH and UCY.** ETH (Pellegrini et al., 2009) and UCY (Lerner et al., 2007) are two common benchmarks for pedestrian trajectory prediction. These datasets consist of 5 subsets of widely used real-world pedestrian trajectories (Alahi et al., 2016; Gupta et al., 2018; Mohamed et al., 2020; Salzmann et al., 2020). The primary challenge in these datasets are the frequent interactions of agents in very crowded scenes. Furthermore, the number of pedestrians varies considerably. Some frames contain only 2 pedestrians, while many have over 50.

**SportVU.** The STATS SportVu (SportVU) is a tracking dataset composed of multiple NBA seasons. Each scene consists of two teams of 5 players, with each team categorized as either making an offensive or defensive play in a particular game. Each play contains 50 timesteps sampled at 5Hz, with the player trajectories expressed in $(x, y, z)$ coordinates.

**Stanford Drone.** The Stanford Drone Dataset (SDD) (Robicquet et al., 2016) is a large dataset with 20 different top-down scenes across multiple areas at Stanford University. Each scene consists of agents of different types, from pedestrians and skaters to cars and buses. Trajectories are recorded at 2.5Hz and expressed in $(x, y)$ world coordinates. Despite the heterogeneity of agents, the maximum number of agents in any one scene is 21.

## B.2 QUALITATIVE RESULTS

In Fig. 5 we plot the node-degree distribution learned by our DGG across multiple datasets. While on average, a pedestrian may only look at their 2 nearest neighbors in crowded scenes such as Zara1 and Univ, this can increase to almost 5 nearest neighbors in some cases. This suggests that both a fully-connected graph, or one with a fixed node-degree like DGM (Kazi et al., 2020) are both suboptimal.

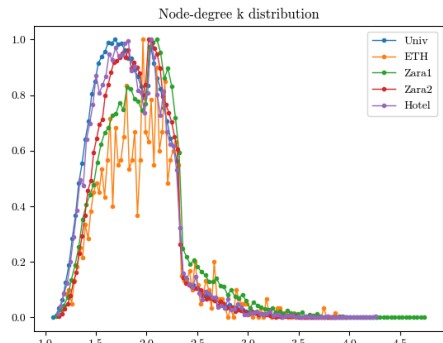

Figure 5: Distribution of the learned node degree $k$ over the test split for different trajectory prediction datasets.

Figure 6 compares our predicted trajectories to DGM (Kazi et al., 2020), on the SportVU dataset. As shown, our trajectories are closer to the ground truth. We illustrate this further in Fig. 7, which shows the graph generated by our DGG for 3 different basketball players in a game. The figure demonstrates how our DGG lets each player look at a different number of neighbours, while DAGNet Monti et al. (2021) forces each player to look at all others in the game.

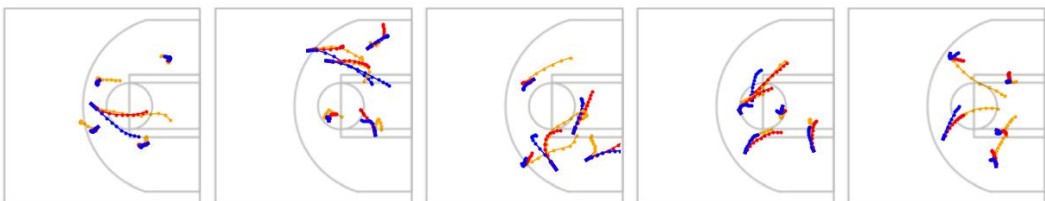

Figure 6: Qualitative results for trajectory prediction on the SportVU dataset. Orange: ground truth; Blue: DGM prediction; Red: our prediction.

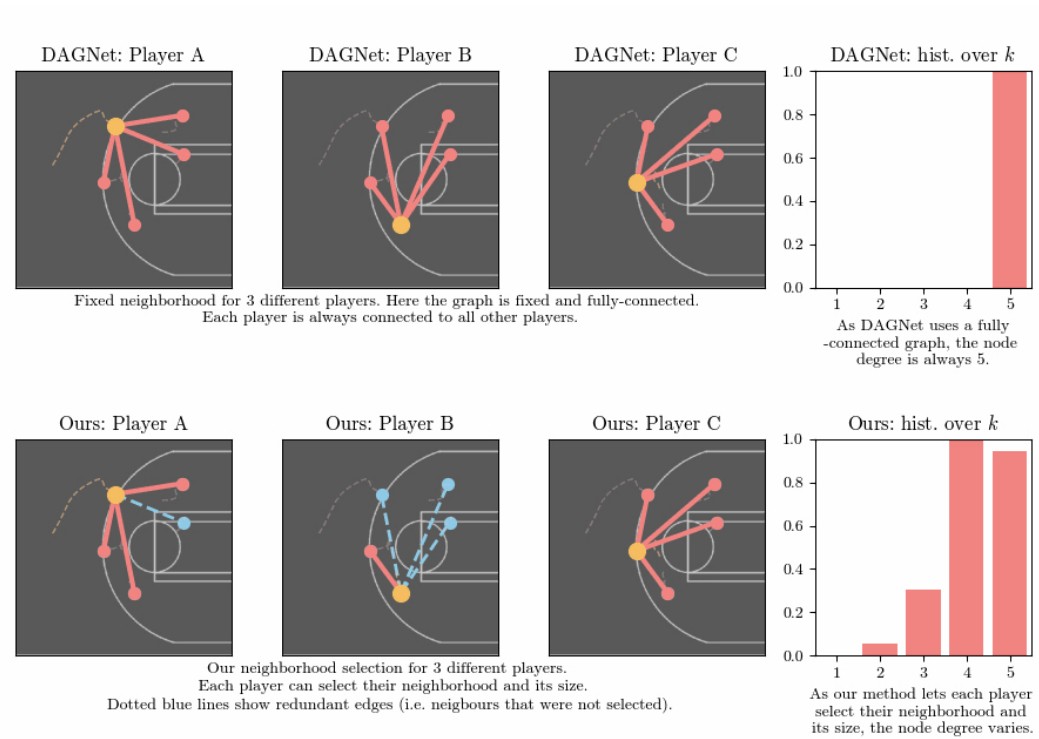

Figure 7: Graph topology visualisation in basketball players on the SportVU dataset: ours vs. DAGNet (Monti et al., 2021). We display the selected neighborhood for 3 different players and a histogram of the node-degree $k$ accumulated over the dataset/scene. First row: DAGNet's fully-connected graph, second row: our DGG.

