# OpenReview forum: "Learning Graph Neural Network Topologies"
_ICLR.cc/2023/Conference — Submitted to ICLR 2023_

### Official Review · Reviewer_CYQs · 2022-10-24

**Confidence:** 4
**Correctness:** 3
**Technical Novelty And Significance:** 3
**Empirical Novelty And Significance:** 3
**Recommendation:** 3

**Clarity, Quality, Novelty And Reproducibility:**

The paper is well-written.

The code and details on training the model is not provided.


**Strength And Weaknesses:**


**Strengths**

The proposed method helps create a similarity graph with a learnable node degree for each node in the graph.

**Weaknesses**

The proposed method does not discuss or compare with a large body of related work for learning a graph topology on the fly (e.g., [1, 2, and 3]). Without comparing with the existing works, the contribution of the proposed method is not measurable.

The time complexity of the proposed approach is not discussed. For the edge ranking step, the similarity of each pair of nodes should be computed and later used in the Gumbel-Softmax. This means O(n**2) time complexity with n as the number of nodes in the graph.

The datasets used in this paper are relatively small compared to the current benchmarks (e.g., OGB graphs [4])

[1] Learning discrete structures for graph neural networks, ICML 2019.

[2] Iterative Deep Graph Learning for Graph Neural Networks: Better and Robust Node Embeddings, NeurIPS 2020.

[3] SLAPS: Self-supervision improves structure learning for graph neural networks, NeurIPS 2021.

[4] Open graph benchmark: Datasets for machine learning on graphs, NeurIPS 2020.



**Summary Of The Paper:**

This paper proposes an end-to-end differentiable graph generator that builds the graph topology simultaneously as learning for a downstream task. The proposed idea extends the applicability of GNNs to cases where a graph topology is not readily available. The experimental results show that the proposed model improves the performance of node classification ad trajectory prediction tasks.

**Summary Of The Review:**

**Questions**

Table 2: It is not clear to me what is used as input to DGG for this experiment. Is the original graph structure for Cora, Citeseer, ... used in this experiment? Also, as mentioned earlier, many baselines are missing here.


The intermediate loss: Why adding the intermediate loss can help? Why doesn't the downstream task loss do the same thing as encouraging nodes in the same class to be connected and nodes in a different class to be disconnected?

---

> ### Author Response · Authors · 2022-11-18
> **Response to reviewer CYQs**
>
> Thank you for your time and feedback.
>
> 1. Differences to related work:
>     - Compared to [1] and [3] our method is a general module that can be integrated into any graph convolution layer. In contrast, those two approaches are entire frameworks with specific training routines.
>     - While [2] is agnostic to the type of GCN, it differs from ours in several ways. Firstly, our method is stochastic and we learn distributions over the adjacency matrix and the node degree. From the distributions we sample the neighborhood for each node and its size. This allows us to automatically select neighborhoods of varying sizes, and learn it in a differentiable manner.
> 2. Results comparison to related work:
>     - One reason we do not compare to [1] and [3] is that their approaches are standalone models instead of integrateable modules. This makes comparison unfair as our results are about demonstrating the effect of our module when integrated into different types of GCNs. Furthermore, the results reported in these two papers uses a different experiment setting to ours: for training they use the training set and half the validation set, while we only use the training set.
>     - We can compare our results to [2] as they present a GCN agnostic method. In fact our integration into a regular GCN outperforms their accuracies on Citeseer and Pubmed, and is 0.6\% less on Cora.
> 3. Time complexity and scalability:
>     - The time complexity is O(n**2) as you correctly stated, which can make scalability an issue. We would like to address this in a future line of work. However we did test on larger datasets such as Reddit, however these are still much smaller than OGB [4].
> 4. Input to DGG for Cora, Citeseer etc in Table 2:
>    - In Table 2 the input to the DGG includes the original graph structure of the dataset.
> 5. Reason for the intermediate loss:
>     - Although the downstream classification loss encourages edges between nodes of the same class and tries to remove those between classes, traversing the loss landscape from the initial random initialization of the network to one where the model is able to discriminate between these nodes can take several iterations using only the downstream loss. The intermediate loss then helps up training, and we anneal it over the training cycle, eventually leaving only the downstream loss.

---

### Official Review · Reviewer_Ds7V · 2022-10-26

**Confidence:** 4
**Correctness:** 3
**Technical Novelty And Significance:** 3
**Empirical Novelty And Significance:** 2
**Recommendation:** 6

**Clarity, Quality, Novelty And Reproducibility:**

The paper is clear, and of relatively high quality. The approach is relatively novel, and adequate details are provided to reproduce the results. I believe that once the authors release the code the work can be easily accessible.

**Details Of Ethics Concerns:**

No specific concerns.

**Strength And Weaknesses:**

Strength: The topic is of interest to the (graph) machine learning community. Updating/learning/rewiring graphs for a final end-task is an open problem with some approaches already proposed. This work builds on those approaches and improves some of those aspects. The paper is very clearly written and easy to follow.

Weaknesses:  I don't see a strong weaknesses of the paper expect from the fact that is it incremental in comparison to the existing literature.

**Summary Of The Paper:**

The paper introduces a method for learning a graph topology that can be integrated in any GNN framework. The proposed probabilistic method is built on a differentiable graph operator that is able to decide the size of the neighbourhood of each node (i.e., degree of each node), as well as the corresponding edges. Experimental results on i) different GNN architectures and ii) a wide set of datasets are provided.

**Summary Of The Review:**

Overall, this is a well-written paper addressing an interesting topic. Although there is quite a lot of work in learning the connectivity matrix in a GNN framework, this paper brings one additional solution to the problem. Thus, the problem itself is not novel, but the proposed solution is valid.

Some additional comments that could hopefully help the authors are the following:

1) In Eq. (8), the dimensionality of $z_i$ and $h_i$ does not seem to be the same. The former is $d$ and the later is 1. Could you please explain and clarify?
2) Can you discuss on the type of edges that you learn? If I understand correctly the learned graph is directed. Is it possible to extend it to an undirected graph, which might be more suitable for some applications?
3) In the training details, the authors mention that "to speed up training, we add an intermediate loss...". Can you please elaborate on that? It seems that the loss promotes a specific class of graphs that consists of community structures. Is that correct? Please clarify.
4) Related to the previous comment, it would be useful to understand more the properties of the learned graph. Can the method be extended to unsupervised settings?

Some examples typos that should be fixed:
- p.7: 'fixe k': remove e
- p.12: 'signle': single

---

> ### Author Response · Authors · 2022-11-18
> **Response to reviewer Ds7V**
>
> Thank you for the favourable assessment and feedback.
>
> 1. Equation 8 clarification: in Eq.8 you are correct that the dimensionality of $z_i$ and $h_i$ are different. In fact we misplaced the brackets, it should be $k_i = D_\rho(z_i) + h_i$. We formulated it this way for two reasons: (1) $h_i$ forms an initial estimate of the node degree which is then updated with $D_\rho(z_i)$ and (2) it ties the representation space of the node degree directly to that of the edge samples (as $h_i$ is the L1-norm of the edge samples).
> 2. Learning an undirected adjacency matrix: the learned adjacency matrix is directed, however it can be made undirected by making the matrix symmetric.
> 3. Intermediate loss clarification: yes, the intermediate loss promotes community structures, where each community represents the nodes for a single class. The idea is that the loss helpes remove edges between classes and only maintains those within a class.
> 4. Application in unsupervised settings: at present our method requires supervision from the downstream task alongside the intermediate loss. This supervision comes from the node labels. In theory, if the community structures of the graph can be obtained purely through the graph topology and without node labels, then these community structures could be used as supervision for the intermediate loss.

---

### Official Review · Reviewer_9Mc3 · 2022-10-26

**Confidence:** 4
**Correctness:** 2
**Technical Novelty And Significance:** 2
**Empirical Novelty And Significance:** 2
**Recommendation:** 3

**Clarity, Quality, Novelty And Reproducibility:**

Clarity:
The paper in general is well organized and easy to follow.

Quality:
The paper is of fair quality in general.

Novelty:
The focused technical aspect of this paper is not novel. There are a few existing works that have already studied joint and differentiable learning of graph topology and node representations (see weakness part for details). Also the proposed method is a combination of techniques that have already been explored by a few existing works (see weakness part for details). In contrast, the differentialable learning of adaptive number of neighbors is somewhat novel.

**Strength And Weaknesses:**

Strengths:

1. The paper aims to optimize a graph model that jointly learns the topology and graph representation in an end-to-end differentiable manner. The cases where adjacency matrix is not provided are supported, as well as refine the given adjacency to improve the topology for better performance

2. A framework is proposed that can plugs in a wide range of existing GNN methods

3. An insight of this work is that it considers the variable number of neighbors in the graph topology learning. The differentiablity is implemented by a combination of degree estimator and top-k neighbor selector.


Weaknesses:

1. An important line of related works are missing. The paper claims that it is the first to consider differentiable end-to-end learning of graph topology, and however, it is wrong. Differentiable graph structure learning has been explored by quite a few existing studies, e.g. [1-5]. In particular, [1] uses bi-level optimization for learning structures/representations as the outer/inner loop, [2, 3] harnesses variational inference for joint learning topology and node representations, [4] considers iterative learning of the two things and [5] proposes a scalable model that learns optimal topology with Gumbel trick in each representation layer. A detailed comparison with these works are needed to illuminate the differences and novelty.

2. The experiment only compares with the methods which drops edges from the orignal graph, instead of the more related and strong competitors that learn the graph structure, especially the works that involve graph topology and representation joint learning, e.g. the above-mentioned models.

3. The proposed method seems to have O(N^2) complexity w.r.t. node numbers and not scalable enough for large datasets. The experiments are only conducted on small datasets, instead of the larger ones, e.g., from OGB.

4. The paper claims that the framework fits with any “graph convolutional networks”, are there any reasons that other spatial GNNs based on message passing do not fit here?

5. The paper uses Gumbel-softmax for edge probabilty estimation. However, the pairwise edge probability is estimated independently of each other. Under this assumption, the dependencies among edges are not taken into account. Are there some jusfitication of this assumption?

[1] Learning Discrete Structures for Graph Neural Networks, ICML19

[2] Variational Inference for Graph Convolutional Networks in the Absence of Graph Data and Adversarial Settings, NeurIPS20

[3] Variational Inference for Training Graph Neural Networks in Low-Data Regime through Joint Structure-Label Estimation, KDD22

[4] Iterative Deep Graph Learning for Graph Neural Networks - Better and Robust Node Embeddings, NeurIPS21

[5] NodeFormer: A Scalable Graph Structure Learning Transformer for Node Classification, NeurIPS22

**Summary Of The Paper:**

In this paper, a joint topology and graph representation learning framework is proposed to tackle the adjacency matrix missing scenarios. The method could also be applied to the given graph topology to improve the emirical performance. The distinguishing point is the incorporation of adapative number of neighbors k which could be differentiably learnt via a Gumbel-softmax trick. However, the paper lacks more detailed discussion and experiment with recent graph structure learning models.

**Summary Of The Review:**

This paper studies an important problem. However, the proposed method lacks enough novelty, especially quite a few proposed components have already been explored by prior art in the same direction. Also, an important line of related works are missing without any comparison, which makes the technical contributions of this work not convincing.

---

> ### Author Response · Authors · 2022-11-18
> **Response to reviewer 9Mc3**
>
> Thank you for your time and thorough comments.
>
> 1. Regarding the rmissing related work:
>     - Broadly, difference between our approach and [1,2,3,5] is that we propose a general topology learning module which can be integrated into any GCN and trained with the downstream task loss. The other approaches are typically networks with specific training schemes.
>     - The variational inference framework used by [2,3] approximates a distribution over the adjacency matrix like we do. However, they sample the entire adjacency matrix while we factorize the distribution to sample each neighborhood independently. Once we sample each neighborhood, we then also induce sparsity on it by learning the node degree $k$ and using it to select the top-$k$ neighbours.
>     - The Gumbel trick used by [5] to sample the edges for each node is performed $k$ times, where $k$ is a hyperparameter. This is where our novelty lies, as we learn $k$ and use it to automatically select each node's neighborhood and its size.
> 2. Experimental comparison to related work:
>     - We only compare to other methods which are integrateable modules. Furthermore, the methods [1-5] use different experiment settings on the benchmark graph datasets. For training they use the training set and half the validation set. We only use the training set.
> 3. Integrating our module into other spatial GNNs:
>     - To learn the adjacency matrix in a differentiable manner our module has to be integrated into a graph convolution operation. Some spatial GNNs that operate on predefined graphs use node indices instead of the adjacency matrix; however, message-passing defined in this manner prevents the edge structure from being learnable.
> 4. The use of pairwise edge probabilities:
>     - Considering dependencies among edges when estimating their probabilities is an excellent suggestion. In our case, given that a node's state (after the first GCN layer) is typically an aggregation of its neighbours, the affinity between two nodes in one layer is tied to the edge structure of the previous layer. We felt this was enough to estimate the probability of an edge, however explicitly taking dependencies between edges into account is a great suggestion for our improvements.

---

### Official Review · Reviewer_ob93 · 2022-10-27

**Confidence:** 4
**Correctness:** 3
**Technical Novelty And Significance:** 2
**Empirical Novelty And Significance:** 2
**Recommendation:** 3

**Clarity, Quality, Novelty And Reproducibility:**

The paper is clearly written. The font of the submission is not standard, though.

**Strength And Weaknesses:**

Strength:
- The paper is clearly written.
- The approach is technically sound.

Weakness:
- The idea of learning the graph structure for GNN is not novel. I have read several papers with similar ideas, for example, http://proceedings.mlr.press/v97/franceschi19a/franceschi19a.pdf
- The proposed method is only evaluated with basic GNN models - the reported number is far below the state-of-the-art performance, on the classic benchmark datasets being used.
- The approach is clearly not going to scale to large graphs. And I did not find discussions regarding the scalability of the method.

**Summary Of The Paper:**

This paper proposes a differentiable graph-generator that builds the graph topology on the fly.


**Summary Of The Review:**

Overall, this paper proposes yet another method for applying Graph Neural Networks to datasets without explicitly relational structure through inferring latent graphs. While the proposed method is technically sound, it lacks technical novelty. Also, the empirical performance is weak, and the method is not scalable.

---

> ### Author Response · Authors · 2022-11-19
> **Response to reviewer ob93**
>
> Thank you for your time and comments.
>
> 1. Novelity:
>     - While there are other methods for learning graph structure for a GCN, the novelty of our method lies in our generation of each node's neighborhood with *automatic size selection*, all in a differentiable manner. Additionally, compared to the paper you suggested, our method is a general module that can be integrated into any graph convolution operation, as opposed to a specific framework. This is one reason we only compare to other methods which are also integrateable modules.
> 2. Choice of GNNs in experiments and resulting performance:
>     - We integrated our module into 4 representative GCN baselines to show its versatility. The aim of our experiments was to show the accuracy improvements that can be brought about by our module. One of the baselines GCNII [1] is state-of-the-art, and our module brings accuracy improvements here too.
> 3. Scalability:
>     - As our approach is $O(n^2)$ you are right that it will struggle with very large graphs such as OCB [2]. Scalability is an excellent suggestion for the direction of our future work.
>
> [1] Simple and Deep Graph Convolutional Networks, ICML 2020.
>
> [2] Open graph benchmark: Datasets for machine learning on graphs, NeurIPS 2020.

---

### Decision · Program_Chairs · 2023-01-20

**Decision:**

Reject

**Justification For Why Not Higher Score:**

Reviewers raised several concerns including limited novelty, issues of scalability to larger graphs, lacking more detailed discussion and experiment with recent graph structure learning models, and consideration of dependencies among edges. The authors’ responses addressed only some of these concerns. Overall, there still seems to be a lack of enough enthusiasm among the reviewers after the author responses.

**Justification For Why Not Lower Score:**

N/A

**Metareview: Summary, Strengths And Weaknesses:**

This paper proposes a differentiable graph-generator that builds the graph topology on the fly. To this end, a joint topology and graph representation learning framework is proposed to tackle the adjacency matrix missing scenarios. The method could also be applied to the given graph topology to improve the empirical performance. The distinguishing point is the incorporation of adaptive number of neighbors k which could be differentiably learnt via a Gumbel-softmax trick. Experimental results on several GNN architectures and multiple datasets are provided and show that the proposed model improves the performance of node classification ad trajectory prediction tasks. Reviewers raised several concerns including limited novelty, issues of scalability to larger graphs, lacking more detailed discussion and experiment with recent graph structure learning models, and consideration of dependencies among edges. The authors’ responses addressed only some of these concerns. Overall, there still seems to be a lack of enough enthusiasm among the reviewers after the author responses.